# Defense of Milkweed Bugs (Heteroptera: Lygaeinae) against Predatory Lacewing Larvae Depends on Structural Differences of Sequestered Cardenolides

**DOI:** 10.3390/insects11080485

**Published:** 2020-07-31

**Authors:** Prayan Pokharel, Marlon Sippel, Andreas Vilcinskas, Georg Petschenka

**Affiliations:** 1Institute of Phytomedicine, University of Hohenheim, 70599 Stuttgart, Germany; prayan.pokharel@uni-hohenheim.de; 2Institute for Insect Biotechnology, Justus Liebig University Giessen, 35392 Giessen, Germany; marlon.sippel@student.uva.nl (M.S.); andreas.vilcinskas@agrar.uni-giessen.de (A.V.)

**Keywords:** predatory–prey interactions, multi-trophic interactions, cardiac glycosides, cardenolides, Lygaeinae, plant toxins, milkweed bugs

## Abstract

Predators and parasitoids regulate insect populations and select defense mechanisms such as the sequestration of plant toxins. Sequestration is common among herbivorous insects, yet how the structural variation of plant toxins affects defenses against predators remains largely unknown. The palearctic milkweed bug *Lygaeus equestris* (Heteroptera: Lygaeinae) was recently shown to sequester cardenolides from *Adonis vernalis* (Ranunculaceae), while its relative *Horvathiolus superbus* also obtains cardenolides but from *Digitalis purpurea* (Plantaginaceae). Remarkably, toxin sequestration protects both species against insectivorous birds, but only *H. superbus* gains protection against predatory lacewing larvae. Here, we used a full factorial design to test whether this difference was mediated by the differences in plant chemistry or by the insect species. We raised both species of milkweed bugs on seeds from both species of host plants and carried out predation assays using the larvae of the lacewing *Chrysoperla carnea*. In addition, we analyzed the toxins sequestered by the bugs via liquid chromatography (HPLC). We found that both insect species gained protection by sequestering cardenolides from *D. purpurea* but not from *A. vernalis*. Since the total amount of toxins stored was not different between the plant species in *H. superbus* and even lower in *L. equestris* from *D. purpurea* compared to *A. vernalis*, the effect is most likely mediated by structural differences of the sequestered toxins. Our findings indicate that predator–prey interactions are highly context-specific and that the host plant choice can affect the levels of protection to various predator types based on structural differences within the same class of chemical compounds.

## 1. Introduction

Top-down regulation by predators is a major force controlling the dynamics of prey populations [1,2]. While many insect species defend themselves against predators with chemical compounds synthesized de novo [3], insects from at least six orders employ secondary metabolites sequestered from their host plants as a defense [4]. Plant secondary metabolites are often toxic or deterrent in order to repel herbivores [5,6] and it has been repeatedly shown that many insect species use sequestered plant toxins for protection against predators. For example, triodine swallowtail butterflies sequester aristolochic acids that are distasteful to birds [7,8], pyrrolizidine alkaloids sequestered by arctiid moths act as defensive agents against *Nephila* spider [9,10], geometrid moths sequestering grayanotoxins are protected against house-lizards [11], and lygaeid bugs and danaine butterflies gain protection against avian predators by sequestering cardenolides [12,13].

Milkweed plants (*Asclepias* spp.) produce cardenolides, and the interactions among milkweed, its specialist herbivorous insects, and predators, represent an important model system in chemical ecology and insect–plant coevolution [14,15,16]. Cardenolides are a group of secondary plant metabolites that are distributed across approximately 62 genera in more than 10 plant families [17,18] occurring in a wide range of habitats [19]. These compounds are an important class of natural drugs, which show cardiotonic and neurological activity in vertebrates and are also toxic for some insects [17,20,21]. The pharmacological effects of cardenolides are mediated by the specific inhibition of the ubiquitous animal enzyme Na^+^/K^+^-ATPase, a cation carrier that is involved in many essential physiological functions such as the generation of neuronal action potentials and the maintenance of an electrochemical gradient across the cell membrane [22,23].

Insects in at least five orders (Diptera: Agromyzidae, Lepidoptera: Danaidae, Coleoptera: Chrysomelidae, Heteroptera: Lygaeidae, Sternorrhyncha: Aphididae, and Caelifera: Pyrgomorphidae) sequester and show adaptations to cope with cardenolides. These species possess a modified form of Na^+^/K^+^-ATPase that is resistant to cardenolides, due to a few amino acid substitutions, a phenomenon referred to as target site insensitivity [15,16,24,25,26,27]. In some cases, the evolution of this trait seems to be associated with the ability to sequester cardenolides for defense [28]. Besides resistance, these insects also show morphological adaptations related to chemical defense mechanisms based on cardenolides. For example, the large milkweed bug *Oncopeltus fasciatus* has evolved a double-layered epidermis to store and deploy cardenolides when attacked by a predator [29,30].

Several studies provide evidence for the distastefulness of plant-derived cardenolide defenses against both vertebrate and invertebrate predators. The most prominent example is the feeding trials involving *Asclepias*-reared monarch butterflies *Danaus plexippus* and blue jays *Cyanocitta cristata*, showing pictures of rejection and disgust behaviors from the birds [31]. Further examples include the mice species *Peromyscus aztecus* and *Reithrodontomys sumichrasti* that taste-rejected both field-caught and laboratory-reared monarchs, as well as diets containing digitoxin, a pharmaceutically important cardenolide from foxglove (*Digitalis* spp.) [32,33]. Similarly, the oleander seedbug *Caenocoris nerii* (Heteroptera: Lygaeinae), reared on cardenolide-containing *Nerium oleander*, was protected against common quails *Coturnix coturnix* [34].

Besides the observations based on vertebrates, effects have also been found for invertebrate predators. Praying mantids (*Tenodera ardifolia*) vomited and showed signs of poisoning after feeding on *O. fasciatus* [35]. The orb-weaving spider *Zygiella x-notata* consumed fewer toxic oleander aphids (*Aphis nerii*) compared to non-toxic aphids. Moreover, spiders built disrupted webs when feeding on toxic aphids [36]. Similarly, *Asclepias* seed-fed adults and nymphs of *O. fasciatus* were significantly less likely to be preyed upon by *Nephila senegalensis* spiders than control bugs raised on sunflower seeds [37]. Even the eggs of milkweed-raised females of *O. fasciatus*, that are known to contain cardenolides via maternal transfer [38], were found to be protected against the larvae of the lacewing *Chrysoperla carnea* [39].

Milkweed bugs (Heteroptera: Lygaeinae) are a diverse group of over 600 species [40] that are typically aposematic, with a distinctive red and black pattern. Across their global distribution range, milkweed bug species are commonly associated with host plants belonging to the family Apocynaceae (e.g., *Asclepias* spp., *Nerium oleander*), which often contain cardenolides [19,26,41]. Several species of milkweed bugs are also associated with plants [42,43,44] that are phylogenetically disparate from Apocynaceae but convergently produce cardenolides [45]. For example, *Lygaeus equestris* (Linnaeus 1758) is associated with the cardenolide containing *Adonis vernalis* (Ranunculaceae) [46,47]. Similarly, *Horvathiolus superbus* (Pollich 1781) specializes on *Digitalis purpurea* (Plantaginaceae) [48,49], which also contains cardenolides [18].

We have demonstrated previously that both, *H. superbus* and *L.equestris*, were protected against avian predation when they had sequestered cardenolides from their respective host plants [45]. However, while the early instar larvae of *H. superbus* raised on *Digitalis* seeds also gained protection against lacewing larvae, *L. equestris* nymphs were not protected when raised on *Adonis* seeds [45]. Since both insect species sequester cardenolides from their respective host plants, it remains unclear which factors mediate these differences. More specifically, predator aversion could be either due to the quantitative or qualitative differences of the cardenolides sequestered. Alternatively, the differential exposure of sequestered defenses to the attacking predator by the two milkweed bug species could explain the observed differences (i.e., the differences could depend on the insect species).

We designed a full factorial experiment to determine how two insect species sequestering the same class of toxic compounds could have such different outcomes with predators. We raised the two milkweed bug species on both types of cardenolide-containing seeds (either *Digitalis* or *Adonis*), and exposed them to lacewing larvae. In addition, we compared the cardenolide profiles of the toxins sequestered by both species, and tested if the amount and identity of the sequestered toxins differed across the two diets and insect species. Specifically, we tested if the different predator tolerance was due to (i) different amounts of cardenolides sequestered from *Digitalis* compared to *Adonis* (i.e., quantitative differences), (ii) the structural differences between the cardenolides sequestered from *Digitalis* and *Adonis* (i.e., qualitative differences), and/or (iii) the differences mediated by the milkweed bug species (e.g., deployment of toxins).

## 2. Materials and Methods

### 2.1. Insect Culture

We collected *Lygaeus equestris* specimens from an *A. vernalis* habitat (“Oderhänge Mallnow”), north of Lebus, Brandenburg, Germany, and the specimens of *Horvathiolus superbus* from a *D. purpurea* habitat close to Eberbach, Baden-Württemberg, Germany. In the laboratory, insect colonies were housed in plastic boxes (19 × 19 × 19 cm) covered with gauze in a controlled environment (Binder KBWF 240) at 28 °C, 60% humidity and a day/night cycle of 16/8 h under artificial light. We reared stock colonies of both species on organic sunflower seeds (Alnatura GmbH, Darmstadt, Germany) and supplied water in Eppendorf tubes plugged with cotton. We also included a piece of cotton as a substrate for oviposition. *H. superbus* used for the video-recording of aversive predator behavior were collected from a *D. purpurea* habitat close to Lollar, Hesse, Germany.

### 2.2. Predation Assay

We obtained *L. equestris* and *H. superbus* eggs from the stock colonies and placed them in Petri dishes (60 × 15 mm, with vents) lined with filter paper. The larvae were either raised on field-collected *Digitalis purpurea* seeds (Eberbach, Germany), commercial *Adonis vernalis* seeds (Jelitto Staudensamen GmbH, Schwarmstedt, Germany), or sunflower seeds as a control, until reaching the second instar (older larvae of *L. equestris* are too big as a prey for lacewing larvae). Water was supplied in Eppendorf tubes as described above. Lacewing larvae (*Chrysoperla carnea*) were obtained commercially (Sautter & Stepper GmbH, Ammerbuch, Germany), transferred individually into the wells of a 48-well plate, supplied with the eggs of *Sitotroga cerealella* (Katz Biotech AG, Baruth, Germany) as a diet and covered with a breathable membrane (Breathe-Easy sealing membrane, Diversified Biotech, Dedham, MA, USA). To increase the body size, lacewing larvae were allowed to feed for two days at 21 °C, 60% humidity and a day/night cycle of 16/8 h under artificial light (Binder KBWF 240 climate chamber, Tuttlingen, Germany). Before the predation experiment, each final instar lacewing larva was transferred into an empty well of a 48-well plate and starved for two days under the same conditions as described above.

Predation assays were carried out under ambient conditions in the laboratory. We exposed the second instar larvae of *L. equestris* and *H. superbus* individually to one lacewing larva in a Petri dish (60 mm diameter) and observed the behavior of the lacewing larva. If the first attack was unsuccessful, i.e., if the lacewing released the milkweed bug instantly after probing, we removed the milkweed bug. These milkweed bug larvae were individually transferred to empty Petri dishes, supplied with sunflower seeds and water, and the survival was scored on the following day. If the attack was successful, we recorded how long the lacewing larvae spent feeding on the milkweed bug larva until the lacewing larva left its prey. We also counted the frequency of aversive behavior (mandible wiping) shown by the lacewing larvae when their attack was unsuccessful. For the illustration of aversive behavior, we recorded the mandible cleaning of lacewing larvae using a camera (Nikon D90, Nikon Corporation, Tokyo, Japan) equipped with a Sigma 105 mm 1:2.8 DG Macro lens (Sigma Corporation, Kanagawa, Japan) in a separate setup.

### 2.3. Chemical Analysis

To analyze the amount and the differences between the sequestered cardenolides in *L. equestris* and *H. superbus*, additional milkweed bug larvae were stored at −80 °C, and subsequently freeze-dried and weighed. The samples were homogenized with zirconia/silica beads (ø 2.3 mm, BioSpec Products, Inc., Bartlesville, OK, USA) in 1 mL HPLC-grade methanol containing 0.01 mg/mL of oleandrin (PhytoLab GmbH & Co. KG, Vestenbergsgreuth, Germany) as an internal standard in a Fast Prep™ homogenizer (MP Biomedicals, LLC, Solon, OH, USA) for two cycles of 45 s at 6.5 m/s. After centrifugation at 16,100 g for 3 min, the supernatant was collected and the sample was extracted two more times with 1 mL of pure methanol. All the supernatants of a sample were pooled and evaporated to dryness under a flow of nitrogen gas. Finally, we dissolved the residues in 100 μL methanol by agitating the samples in the Fast Prep™ homogenizer without the inclusion of beads and filtered samples into HPLC vials using Rotilabo^®^ syringe filters (nylon, pore size: 0.45 μm, diameter: 13 mm, Carl Roth GmbH & Co. KG, Karlsruhe, Germany). 

We injected 15 μL of extract into an Agilent 1100 series HPLC (Agilent Technologies, Santa Clara, CA, USA) equipped with a photodiode array detector and separated the compounds on an EC 150/4.6 NUCLEODUR^®^ C18 Gravity column (3 µm, 150 mm × 4.6 mm, Macherey-Nagel, Düren, Germany). Cardenolides were eluted at a constant flow rate of 0.7 mL/min at 30 °C using the following acetonitrile–water gradient: 0–2 min 16% acetonitrile, 25 min 70% acetonitrile, 30 min 95% acetonitrile, 35 min 95% acetonitrile, 37 min 16% acetonitrile, 10 min reconditioning at 16% acetonitrile. Peaks with symmetrical absorption maxima between 218 and 222 nm were recorded as cardenolides [50] using the Agilent ChemStation software (B.04.03). Finally, we estimated the amount of cardenolides per sample by comparing the sum of all cardenolide peak areas to the area of the internal standard [51,52].

### 2.4. Statistical Analysis

We tested the hypothesis that the diet of the bugs affected their survival upon attack by a lacewing larva using the 2 × 3 Freeman–Halton extension of Fisher’s exact test [53]. The probability values for the binomial data from the predation experiment (*survival of milkweed bugs and mandible cleaning behavior by lacewing larvae*) were computed using an online statistical tool (http://www.danielsoper.com/statcalc) [54]. All the other data were analyzed using the JMP^®^ 14.3.0 statistical software (SAS Institute, Cary, NC, USA). Data were assessed for normal distribution by visual inspection of the q-q plots and by the Shapiro–Wilk W test. Homogeneity of variances was evaluated by visual inspection of residual plots. The duration data from the predation experiment were log_10_ transformed to achieve normal distribution and analyzed using Welch’s ANOVA due to the heteroscedasticity of this dataset. We excluded one outlier (determined by the outlier box-plot in JMP) from the dataset of *L. equestris* raised on *Adonis* but the exclusion of this outlier did not change the direction of results. To assess the differences between treatments, we used the Games–Howell HSD post-hoc test. The concentrations and diversity of sequestered toxins were analyzed by ANOVA followed by LSMeans Difference Tukey HSD. We included bug species, treatment (*Digitalis* or *Adonis*), and the interaction between bug species and treatment in our model. Probability values < 0.05 were considered statistically significant.

## 3. Results

### 3.1. Predation Assay

We conducted predation trials with *H. superbus* and *L. equestris* larvae raised on either sunflower, *A. vernalis*, or *D. purpurea* seeds and the larvae of the predatory lacewing *C. carnea*. We found that a diet of *Digitalis* seeds increased the survival of both *H. superbus* and *L. equestris* (*p* < 0.001, for both insect species, Fisher’s exact test) (Figure 1). In addition, the lacewing larvae showed mandible-cleaning behavior (Appendix A) only after attacking *Digitalis*-raised *H. superbus* and *L. equestris* (*p* < 0.001, for both insect species, Fisher’s exact test) (Figure 2). In contrast, the bugs raised on both *Adonis* (although containing cardenolides) and sunflower seeds were neither protected, nor did the lacewing larvae show mandible-cleaning behavior after attacking them (Appendix A). Moreover, lacewing larvae spent significantly less time feeding on both *H. superbus* and *L. equestris* raised on *Digitalis* seeds as compared to *Adonis* and sunflower-raised bugs (*p* < 0.001, for both insect species, Games–Howell HSD) (Figure 3).

### 3.2. Chemical Analysis

We assessed the quantity and compared the retention times of the sequestered cardenolides in both species of bugs, raised on *Digitalis* or *Adonis* seeds (Figure 4 and Figure 5). We found an effect of diet on sequestration (F_3,33_ = 3.939; *p* = 0.025, LSMeans Differences Tukey HSD, Figure 4). *Digitalis*-raised *L. equestris* sequestered lower concentrations of cardenolides than the *Adonis*-raised *L. equestris* (*p* = 0.021, LSMeans Differences Tukey HSD), whereas *H. superbus* sequestered similar concentrations of cardenolides from both types of seeds (*p* = 0.998, LSMeans Differences Tukey HSD). Regarding the diversity of sequestered cardenolides, the bugs sequestered fewer structurally different cardenolides (based on retention times) from the seeds of *Digitalis* than from the seeds of *Adonis* (F_3,33_ = 27.623; *p* < 0.001, LSMeans Differences Tukey HSD, Figure 4). Specifically, *L. equestris* sequestered three times more different cardenolides from *Adonis* compared to *Digitalis*. We found the same pattern for *H. superbus* although the difference was less pronounced (*p* < 0.001, LSMeans Differences Tukey HSD). 

### 3.3. Figures, Tables and Schemes

## 4. Discussion

It is widely accepted that sequestered phytochemicals protect herbivorous insects against their natural enemies [12]. However, our understanding of how the structural differences of sequestered plant compounds, either within the same or across different classes of substances, can affect the outcome of predator–prey interactions, is very limited. We showed that the chance of a milkweed bug to survive a lacewing attack strongly depends on the original source of the sequestered cardenolides. Although the milkweed bugs sequestered cardenolides from both the toxic plant species (*A. vernalis* and *D. purpurea*) tested, only the bugs feeding on *Digitalis* seeds gained protection against lacewing larvae. Furthermore, our observations indicate that rejection is based on taste, as we observed aversive behavior (i.e., mandible cleaning) in the lacewing larvae after attacking bugs raised on *Digitalis* seeds. Accordingly, the lacewing larvae spent less time feeding on bugs raised on *Digitalis* seeds than on those raised on *Adonis* or sunflower seeds, indicating that the *Digitalis*-derived cardenolides show deterrent activity, but that the *Adonis*-derived cardenolides did not. Since the bugs sequestered similar or lower amounts of toxins from *Digitalis* compared to *Adonis*, it is very likely that structural features specific to one or more *Digitalis* cardenolides, rather than quantitative differences, increased the survival of milkweed bug larvae. While only cardenolides sequestered from *D. purpurea* increased survival in milkweed bugs, cardenolides from *A. vernalis* also exert some deterrent activity and decrease consumption by lacewing larvae [45]. The pattern observed here was identical for both species of milkweed bugs, rejecting the hypothesis that our initial observation was mediated by features specific to the insect species such as deployment or the discharge of toxins.

Lacewings have been used as natural predators for biological control for more than 250 years [55,56], and *C. carnea* has been employed commercially against various insect pests including lepidopterans [57,58], Colorado potato beetle [59,60], and others [61]. Lacewing prey consumption behavior was reviewed by Principi & Canard [62] and Canard & Duelli [63], and the sequence of attack was described in detail by McEwen et al. [64]. Lacewing larvae recognize their prey by contacting them with their palpi and/or antennae, followed by probing them with their mandibles for chemosensory recognition. Finally, they capture their prey and inject salivary secretions from venom glands at the base of their maxillae [65] via the mandibles, causing prey tissue to liquify, and subsequently draw it up. In our experiments, we found that lacewings rejected apparently distasteful prey, followed by mandible cleaning behavior. We hypothesized that this grooming behavior, to our knowledge reported here for the first time, removes prey toxins by rubbing mouthparts together and wiping them on the substrate.

Lacewing larvae have been reported to acquire resistance against several different pesticides, such as flonicamid (pyridines) [66] and lambda-cyhalothrin (pyrethroids) [67]. However, only a few experimental studies investigated the effects of plant toxins in herbivore diets on lacewings. One such study found that *C. carnea* larvae did not experience increased mortality when they attacked diamondback moths *Plutella xylostella* feeding on plants with toxic glucosinolates or without glucosinolates [68]. However, as mentioned above, cardenolides conferred protection to the eggs of *Asclepias*-raised *O. fasciatus* against *C. carnea* [39]. Surprisingly, cardenolides were found to be present in lacewing pupae when the larvae preyed upon *Aphis nerii* feeding on *Nerium oleander* [69] indicating an uptake of sequestered compounds from the prey by the predator. While the aforementioned examples suggest that lacewing larvae can tolerate insecticides or sequestered plant toxins to some extent, our knowledge on the aversive properties of sequestered plant toxins inducing responses such as the mandible cleaning that we described here seems to be quite limited.

Plants produce a great diversity of secondary metabolites across but also within compound classes and even on the level of individual plants. One potential hypothesis to explain the ecological significance of the diversity of observed chemical defenses is the screening hypothesis [70]. This hypothesis posits that the diversity of plant toxins is sustained to enhance the probability of a plant to possess an effective compound or a precursor of it against multiple predators or combinations of compounds working synergistically [71], thus together generating a selective advantage against a wide range of antagonists. Substantial variation of gross cardenolide content has been reported in natural populations of monarch butterflies [72] and the milkweed bugs *O. fasciatus* and *Lygaeus kalmii* [73]. In monarch butterflies, palatability to blue jays was found to vary depending on the species of milkweed used as a host plant by the caterpillar [14]. Different species of *Asclepias* plants produce structurally diverse cardenolides that differ in their emetic potency against predators [74]. For example, monarch butterflies raised on *A. eriocarpa* had greater emetic potency than monarchs reared on *A. speciosa* [75]. Despite the fact that sequestration in milkweed bugs is comparatively well studied, the potential effects of the quantitative and structural variation of dietary cardenolides in milkweed bugs against their predators have not yet been tested empirically.

Our experiment sought to examine which factors mediated the different outcome of predator–prey interactions in closely related insect species sequestering the same class of compounds from their respective host plants. While *H. superbus* sequestered similar amounts of cardenolides from both plant species, *L. equestris* accumulated lower concentrations of cardenolides from *Digitalis* than from *Adonis*. This suggests that the observed defensive activity of cardenolides obtained from *Digitalis* was not mediated by dose, but rather by structural differences between *Digitalis*- and *Adonis*-derived cardenolides. In line with this, we observed noticeable differences in the structural diversity and polarity of sequestered cardenolides from these two plant species in both species of insects. For both insect species, we found that *Digitalis*-raised bugs sequestered fewer structurally different cardenolides than *Adonis*-raised bugs. Cardenolides sequestered from *Adonis* covered a wider polarity range than the cardenolides sequestered from *Digitalis*. Furthermore, the *Digitalis*-raised individuals of either milkweed bug species sequestered a higher proportion of apolar cardenolide compounds, potentially mediating the observed effect [76]. Here, we did not determine the identities of the individual cardenolides observed. In a previous study [45], the comparison of nine authentic cardenolides from *D. purpurea* with the cardenolides sequestered by *H. superbus* from the same plant revealed no matches based on retention times. For *L. equestris* raised on *A. vernalis* seeds, we tentatively identified cymarin, strophanthin, and k-strophantoside [45]. Although the structural identity of most cardenolides remains unknown, our findings support the hypothesis that individual plant compounds within the same chemical class can act against antagonists selectively.

Predator diversity is probably an important evolutionary driver for the observed wide variation in anti-predator defenses, as different predator species have varying tolerances to toxins, sensory abilities, and attack strategies [77]. Predators as taxonomically diverse as birds and invertebrates may exert differential selection pressures on the same prey [78]. Autogenous production as well as sequestering chemical defenses can incur physiological costs, as the organisms must tolerate active phytochemicals and sometimes modify them, whereas autogenous chemical defenses burden the species’ limited resources at the expense of other functions, such as growth and survival [79,80,81,82]. These costly defenses are effective, but may only evolve to be deterrent against a wide array of natural enemies if required by predation pressure [83]. Unfortunately, our knowledge about predators attacking milkweed bugs in the field is very limited, but maybe lacewing larvae are not preying upon *L. equestris* in *A. vernalis* habitats, and therefore no selection on defenses against lacewings occurred in this species. Notably, *L. equestris* occasionally also uses *D. purpurea* as a host, suggesting that the individuals of this species show variation with regard to the predators they are protected from. Although we did not study predation on milkweed bugs in the field, this suggests that our findings also possess relevance in the field.

Although earlier literature suggested that the cardiotonic activity of *Adonis* and *Digitalis* extracts were equally potent on isolated frog hearts [84], we found that cardenolides from both plant species were perceived differently by predators. This agrees with previous work outlining how various predators reacted differently to the same prey. For example, *Neacoryphus bicrucis*, sequestering pyrrolizidine alkaloids from *Senecio*, were distasteful to green-anole lizards, however, the bugs were palatable to Fowler’s toad [85]. Recently, it was also shown that two different defensive fluids from the thoracic glands and abdomen of the wood tiger moth *Arctia plantaginis* are predator specific. The moth thoracic fluids were deterrent to birds but not ants, and in contrast, the abdominal fluids deterred ants, but birds did not show any aversive response [86]. Besides the differences that may be mediated by host plant quality, the different outcomes of predator–prey interactions can also be mediated by traits of the insects. Although *L. equestris* and *Tropidothorax leucopterus* were both feeding on cardenolide-free *Vincetoxicum hirundinaria*, only *L. equestris* was shown to be defended against domestic chicks [87]. This suggests that only *L. equestris*, but not the closely related *T. leucopterus*, was able to derive a defensive principle from this host plant. Regarding the huge structural diversity of sequestered plant secondary metabolites, future work should focus on the structure–activity relationships in the framework of predator–prey interactions.

## 5. Conclusions

Our results provide evidence that structural differences within the same class of sequestered host-plant toxins can direct the outcome of predator–prey interactions. Our findings indicate that predator–prey interactions are highly context-specific, and that investigating the effects of the diversity of chemical defenses on different predators in a community is vital for understanding tri-trophic interactions within an ecosystem.

## Figures and Tables

**Figure 1 insects-11-00485-f001:**
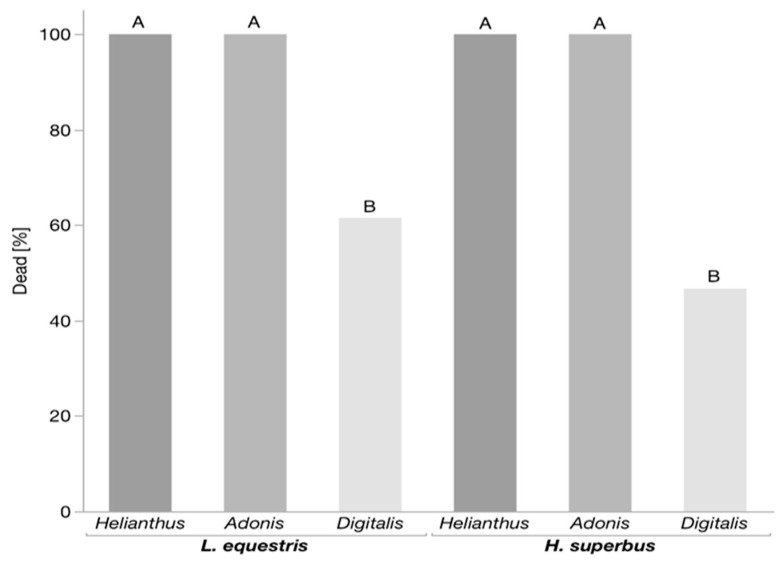
Survival of the milkweed bug larvae preyed upon by *C. carnea*. Bars represent the proportion of dead larvae after *C. carnea* attacks. The milkweed bugs *L. equestris* (n = 22–26 per treatment) and *H. superbus* (n = 15–16 per treatment), were raised on the seeds of either *Helianthus annus* (sunflower), *Adonis vernalis*, or *Digitalis purpurea*. Within the same insect species, levels not connected by the same letter are significantly different.

**Figure 2 insects-11-00485-f002:**
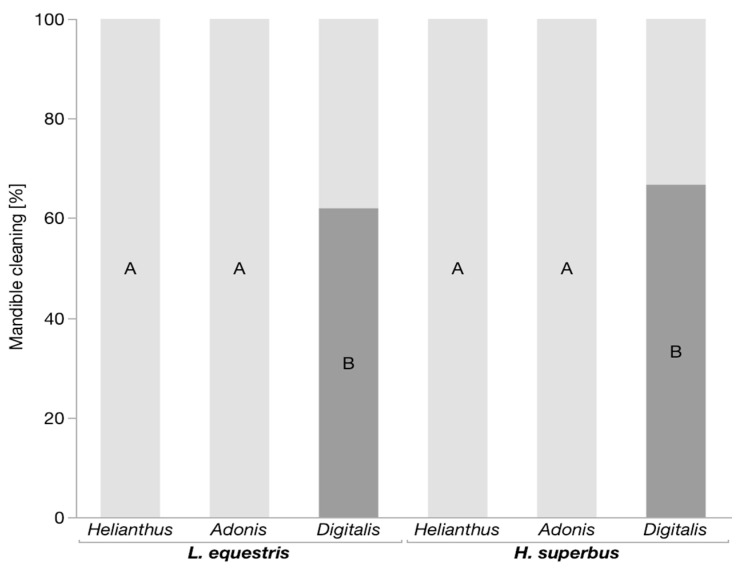
Mandible cleaning behavior shown by *C. carnea* after attacking milkweed bug larvae. Bars represent the proportion of lacewing larvae that cleaned their mandibles (dark grey) and that did not clean their mandibles (light grey) after attacking *L. equestris* (n = 22–26 per treatment) and *H. superbus* (n = 15–16 per treatment) larvae raised on the seeds of either *Helianthus annus* (sunflower), *Adonis vernalis* or *Digitalis purpurea*. Within the same insect species, levels not connected by the same letter are significantly different.

**Figure 3 insects-11-00485-f003:**
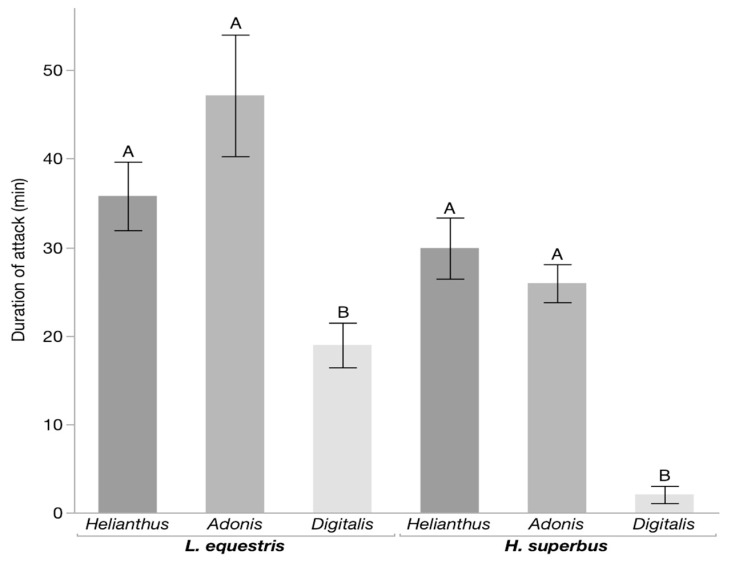
Feeding duration of *C. carnea* on milkweed bug larvae. Bars represent the means ± SE of the time taken by *C. carnea* to feed upon *L. equestris* (n = 22–26 per treatment) and *H. superbus* (n = 15–16 per treatment) larvae raised on the seeds of either *Helianthus annus* (sunflower), *Adonis vernalis*, or *Digitalis purpurea*. Within the same insect species, different letters above bars indicate significant differences.

**Figure 4 insects-11-00485-f004:**
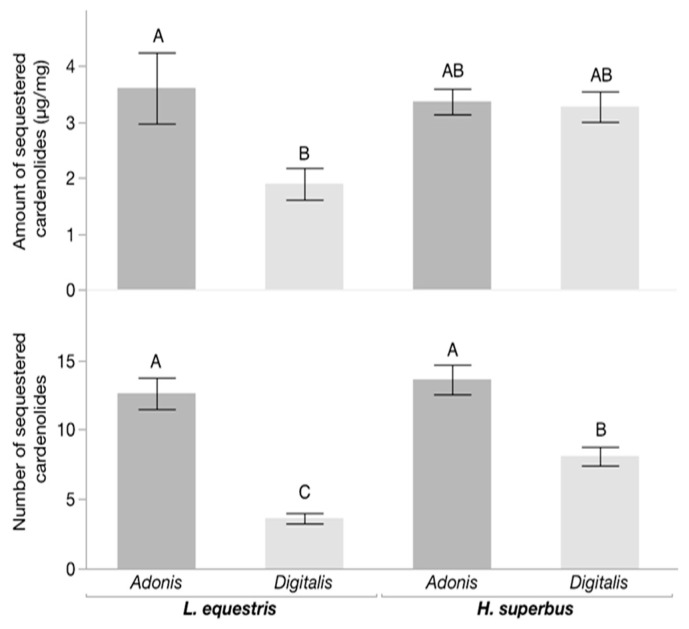
Quantity and number of different sequestered cardenolides by milkweed bug larvae. Bars represent the means ± SE of the concentration (µg/mg) (**top**) and the number of structurally different cardenolides sequestered (**bottom**) by *L. equestris* (n = 8 per treatment) and *H. superbus* (n = 8–10 per treatment) raised on the seeds of either *Adonis vernalis* or *Digitalis purpurea*. Different letters above bars indicate significant differences.

**Figure 5 insects-11-00485-f005:**
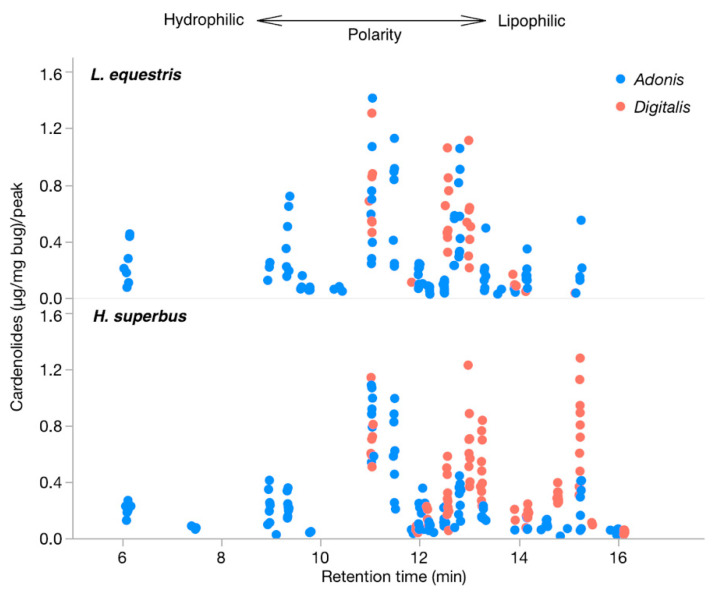
Concentrations of the sequestered individual cardenolides and retention times based on HPLC analysis. Each datapoint represents an individual cardenolide sequestered by a specimen of *L. equestris* (n = 8 per treatment) (**top**) and *H. superbus* (n = 8–10 per treatment) (**bottom**) raised on the seeds of either *Adonis vernalis* (blue) or *Digitalis purpurea* (red). Polar cardenolides (hydrophilic) have shorter retention times and apolar cardenolides (lipophilic) have longer retention times.

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
