# Peer review of "Defense of Milkweed Bugs (Heteroptera: Lygaeinae) against Predatory Lacewing Larvae Depends on Structural Differences of Sequestered Cardenolides"

_insects, 2020, doi:10.3390/insects11080485_

Round 1
Reviewer 1 Report
This study examined the effect of two different diets on the chemical defenses of two aposematic true bugs. The authors found that host plant affected the bugs defenses against lacewing larvae, and that this difference appeared to be the result of differences in the types of cardenolides sequestered from the plants, and not the total amount. This is an interesting and well-written manuscript and I very much enjoyed reading it. I think it makes a useful contribution to the existing literature on chemical defense and predator perception.
I have only one small suggestion. Were you able to identify the different individual cardenolides shown in figure 5? If so I would recommend including these identities either in the main text or supplementary materials.
Author Response
Reviewer 1:
Thank you very much for the positive feedback on our paper! The method of analysis which we used (HPLC-DAD) does not allow for structural identification of individual cardenolides. We counted all peaks that had a typical cardenolide absorption spectrum as a cardenolide, an approach which is widely used by several others in the field (see https://doi.org/10.1007/s10886-019-01055-7, https://doi.org/10.1007/s10886-009-9719-0). We added additional references to the methods section (line 178). We have tried to identify individual cardenolides sequestered by Horvathiolus superbus from Digitalis via retention time using nine reference compounds known to occur in Digitalis purpurea in a previous study (see https://doi.org/10.1101/2020.06.16.150730). Surprisingly, none of the compounds matched what we found in the bugs. In L. equestris, we tentatively identified three compounds. We included these findings and more explanation in the discussion (lines 394-400). We are working on the detailed chemistry of cardenolides sequestered by milkweed bugs, but we have no results to present in this paper, yet.
Reviewer 2 Report
The manuscript by Pokharel et al. seeks to understand the primary factors governing predator protection (or lack thereof) in two closely related milkweed bugs known to sequester cardenolides for different host plants. While both have been shown to gain protection from birds, despite different hosts, only one species (that feeds on D. purpurea) was protected from predatory lacewing larvae. With this in mind, the authors designed a thoughtful experiment coupling bioassays with HPLC to determine if difference in lacewing predation was due to host plant chemistry or inherent features belonging to the insects themselves. Both milkweed bugs exhibited greater survival when feeding on D. purpurea despite the total sequestered cardenolide levels being the same or lower than those sequestered from A. vernalis with fewer total compound sequestered as well. The authors argue that this is evidence that structural diversity via host plant chemistry is responsible for deterrence properties even when sequestering the the same class of compounds. I find this topic to be incredibly interesting from both an evolutionary and ecological perspective, especially given the work they have already done in the system. I also appreciated that the paper was very well-written and it was clear that the authors have spent a lot of time crafting this paper. I also found the mandible cleaning behavior that they reported to be an interesting variable they considered in their analyses.
The main concerns I have are three-fold: 1) the lack of detailed methodology, 2) the absence of chemical diversity data and confusion in what that really means, and 3) the lack of ecological significance. Both the bioassays and the chemical analyses were deficient in a number of critical details and I have outlined specific questions below, but this is certainly not exhaustive. More importantly, considering this study concludes that structural differences in host plant (and thus, sequestered) cardenolides are critical in the predator-prey interactions of interest, structural diversity metrics were notably absent in the chemical analyses and I wonder if more analytical analyses are needed (but perhaps some concerns can be addressed with more details in this section?). It was unclear how the compounds were identified (standards?) and the quantification was all based on an internal standard making the concentrations only relative without the use of commercial standards. This makes me question whether the total amount is accurate or if certain compounds may be under or overestimated using the internal standard only approach. Furthermore, the number of sequestered cardenolides does not directly translate to “structural diversity” and what this term really means was never clearly described. Are you referring to different individual compounds or cardenolides with particular functional groups? Regardless, it is possible that there are a few compounds present in Digitalis that act synergistically to effectively deter lacewing larvae, but if that’s the case, it’s not really the diversity that is offering protection, rather, the presence of those particular compounds. Again, the definition or how the reader is supposed to think about “diversity” is not fleshed out very well. Figure 5 does little to offer insights into what compounds might be responsible, do you have data that can bolster this aspect of the paper, which seems to be the most important? Finally, it was not clear to me if there was some ecological angle that should be considered in this system. While lacewings as biocontrols was mentioned, is it a common predator for both species? It also wasn't clear how likely is it that these species feed on the two host plants tested (only one plant was mentioned for each insect)? Some more ecological context may be helpful.
Specific comments:
L121: Do you have any consumption data for the larvae reared on the different seed types? What is known about the original concentrations and chemical composition of the seeds? Is it possible that the chemical make-up of the sequesterd compounds is reliant upon the host seeds and how much is ingested over time?
L132: Is there a reason 2nd instars were used? Is there information on how levels of cardenolides change in these species with ontogeny?
L134: How long were the trials? What were the sample sizes? Were they recorded? What were the conditions? More information is needed here. Also, the separate set-up to record aversive lacewing behavior is confusing. Was this performed in addition to the original design or completely different set-up? Again, how long were the trials?
L145: “…were killed by freezing, freeze-dried, and weighed” is awkward, please revise.
L148: Two rounds of what? Extractions?
L161: How did you determine the specific identities of the cardenolides? Standards? Retention times?
L162: Can you sum all the cardenolide peaks this way?
L168: Please list the online statistical tool here.
L173: How were outliers determined?
Figure 1: Is there a reason this figures is not a simple bar graph? The light and dark grey bars on top of one another are confusing and not necessary given that if you’re reporting %survival the %dead is easily calculated. Also, were there any instances in which the insects were not attacked by the lacewing larvae?
L266-275: I’m not entirely sure why this paragraph is here because the sequestration potential of the lacewing was not investigated in this study.L313: Can you please explain this reasoning? Is this statement suggesting that insects that feed on Adonis are only protected against birds but not lacewing larvae because birds are their primary predator and thus the compounds they sequester are mostly deleterious to them?
Author Response
Reviewer 2:
Thank you very much for your comments and your thorough review! Please see below for a detailed list of our responses.
L121: Do you have any consumption data for the larvae reared on the different seed types?
No, we did not collect consumption data during this study. However, we presented consumption data for larvae of H. superbus raised on D. purpurea and for larvae of L. equestris raised on A. vernalis in another manuscript that we submitted recently (see https://doi.org/10.1101/2020.06.16.150730). These data revealed that in both systems (H. superbus on D. purpurea and L. equestris on A. vernalis) consumption by lacewing larvae was impaired when feeding on milkweed bug larvae sequestering cardenolides. We thank the reviewer for pointing this out and discussed these findings in the manuscript (lines 288-290).
What is known about the original concentrations and chemical composition of the seeds?
In the same study, we also collected some data on the concentration (and composition) of cardenolides in Digitalis and Adonis seeds but we don’t see how this information relates to our observations reported here.
Is it possible that the chemical make-up of the sequesterd compounds is reliant upon the host seeds and how much is ingested over time?
Of course, it is an option that sequestered cardenolides will change over time and that this will affect lacewings. Nevertheless, milkweed bug larvae of the same species were raised on the seeds until reaching the 2nd instar, i.e. for a similar number of days for each species (lines 125-126) suggesting that our findings are indeed mediated by ‘plant’-differences. If these differences would change over time (e.g. if Adonis-raised bugs would become unpalatable and Digitalis-raised once would become palatable) can’t be answered at present.
L132: Is there a reason 2nd instars were used? Is there information on how levels of cardenolides change in these species with ontogeny?
We used 2nd instars because older larvae are too big for consumption by lacewing larvae (line 126) Unfortunately, we don’t have any information on how the level of cardenolides changes across larval stages in milkweed bugs. Since older larvae become too big as a potential prey for lacewing larvae, we furthermore assume that the ecological relevance of a potential effect of time may not be relevant in this system.
L134: How long were the trials? What were the sample sizes? Were they recorded? What were the conditions? More information is needed here. Also, the separate set-up to record aversive lacewing behavior is confusing. Was this performed in addition to the original design or completely different set-up? Again, how long were the trials?
How long were the trials?
We did not record how long it took until the lacewing larvae attacked since we defined the first (and only attack) as the start of the experiment. If attacks were unsuccessful (lacewings shying back) we removed bugs and recorded survival at the next day. After successful attacks (lacewings started feeding) we recorded the time spent feeding (see below). Our procedure is detailed in the methods section (lines 137-143). After successful attacks, we measured the time taken by lacewing larvae to feed on milkweed bugs until the bug was entirely consumed and the lacewing larva left its prey (lines 140-142). In the dataset shown in Fig. 3 all trials were included (also unsuccessful ones that only took a few seconds, i.e. probing by the lacewing and shying back). During the revision, we realized that it would be better to only include successful trials (i.e. trials in which actual consumption happened) as feeding may be different from probing only (that only occurred in lacewings preying upon Digitalis-raised bugs). We revised figure 3 (line 252) and the analyses accordingly and only included trials during which feeding occurred.
What were the sample sizes?
Sample sizes are mentioned in the figure legends.
Were they recorded?
No, we observed the trials by eye but did not film them. At the same time, we counted the occurrence of mandible cleaning behaviour. Since we are writing ‘observed’ and did not mention any equipment, we hope this should be clear. To avoid confusion, we added more explanation to the part on our recording of mandible cleaning behaviour (see below).
What were the conditions?
We carried out all the predation trials under ambient laboratory conditions and added more technical details on our predation assays to the methods section (line 135).
Also, the separate set-up to record aversive lacewing behavior is confusing. Was this performed in addition to the original design or completely different set-up? Again, how long were the trials?
A few video recordings were carried out in a separate setup to illustrate mandible cleaning behaviour (lines 143-146). Trials in which mandible cleaning behaviour occurred only lasted for a few seconds since they were unsuccessful trials, i.e. lacewings shied back (lines 142-143).
L145: “…were killed by freezing, freeze-dried, and weighed” is awkward, please revise.
We changed our wording to: “… were stored at -80°C, and subsequently freeze-dried and weighed.” (lines 159-160)
L148: Two rounds of what? Extractions?
We changed ‘rounds’ to ‘cycles’ and restructured the sentence, it should be clear now (lines 162-163).
L161: How did you determine the specific identities of the cardenolides? Standards? Retention times?
We did not determine the identities of the individual cardenolides observed. We tried comparisons to commercial standards during a related project (see response to reviewer 1) but without success for H. superbus raised on D. purpurea. We speculate that the cardenolides sequestered by the insects might even represent cardenolides that were not described, yet, and structural identifications are part of our ongoing research. For L. equestris raised on A. vernalis seeds we tentatively identified cymarin, strophanthin, and k-strophantoside based on retention times of authentic standards during the same project. We added this information to the discussion (lines 391-397). Nevertheless, we don’t think that it is critical to know individual cardenolide identities for the conclusions of our paper.
L162: Can you sum all the cardenolide peaks this way?
Cardenolides were identified based on their typical absorption spectra (a single symmetrical peak with a maximum between 218-220 nm). Since all cardenolides share the same chromophore (the unsaturated lactone ring) we may assume that all cardenolides absorb with the same intensity independent of the specific structure of their steroid core and the sugars attached. In addition, there is an extensive body of literature relying on this approach, indicating that it is widely accepted. We added a reference to our methods (line 178). While this does not necessarily mean that this approach is valid, and by far not all cardenolides have been compared, there is no reason to assume that it is not based on the chemistry. In addition, we recently published a paper showing that there is a clear correlation between our estimates and a different approach to quantify cardenolides (based on inhibition of Na+/K+-ATPase, see Züst et al. 2019, Journal of Chemical Ecology, https://doi.org/10.1007/s10886-018-1040-3).
L168: Please list the online statistical tool here.
We added the link and a reference to our manuscript (line 183-184).
L173: How were outliers determined?
Outliers were identified by the outlier box plot in JMP. Whenever outliers were excluded, we reran the analysis and found the same direction of results. We added this information to the methods (lines 188-189). After reanalyzing our dataset on duration of feeding, we excluded only one outlier.
Figure 1: Is there a reason this figures is not a simple bar graph? The light and dark grey bars on top of one another are confusing and not necessary given that if you’re reporting %survival the %dead is easily calculated. Also, were there any instances in which the insects were not attacked by the lacewing larvae?
We agree with the reviewer and changed our design for Figure 1. Now, only the % of dead individuals are plotted in a simple bar chart. We adapted the figure legend accordingly (lines 232-237).
There were no instances that the insects were not attacked by the lacewing larvae. We waited patiently, until the predator attacked the insect.
L266-275: I’m not entirely sure why this paragraph is here because the sequestration potential of the lacewing was not investigated in this study.
This paragraph (lines 305-316) illustrates that lacewings can possess resistance to insecticidal toxins. Moreover, a few studies suggest that they can also tolerate sequestered plant toxins and even ingest plant toxins sequestered by the prey. However, there is limited knowledge on to what extent lacewing larvae are deterred by plant toxins and we found it valid to connect knowledge about resistance/tolerance to our observations on deterrence (mandible wiping).
L313: Can you please explain this reasoning? Is this statement suggesting that insects that feed on Adonis are only protected against birds but not lacewing larvae because birds are their primary predator and thus the compounds they sequester are mostly deleterious to them?
Yes, that’s basically what we mean. Unfortunately, our knowledge on natural predators of milkweed bugs is very limited. We tried to clarify in our manuscript (lines 406-412).
General comments:
Throughout the manuscript, we adapted our wording to clarify that we did not assess the effect of structural diversity (see changes) and we apologize for any confusion. Our study demonstrates that the observed defensive activity of cardenolides obtained from one plant is not mediated by dose (quantity), but rather by structural differences between the cardenolides (quality) between the two different plants (lines 220-230). As outlined above, we did not identify the specific cardenolides in this study. Figure 5 shows that the cardenolide peaks from the two host plants differ from each other differ based on their retention times (i.e. they are structurally different). Unfortunately, our knowledge on milkweed bugs’ predators in the field is extremely limited (lines 406-408) and we are going to conduct research to fill this gap. This means, that we cannot really state how relevant our findings are for predator-prey interactions that actually occur in the field. Nevertheless, since lacewing larvae are common generalist predators it is plausible that they will be preying upon milkweed bug larvae in the field (probably depending on the habitat). L. equestris was also observed feeding on D. purpurea, so in this species it is likely that differentially protected individuals actually occur in the field. We added this information to the discussion (lines 409-412).
We hope that we have sufficiently addressed all suggestions made to improve the quality of our manuscript.
Best wishes,
Georg Petschenka on behalf of all authors